# Hybrid coating prepared with PMMA/Ti-O-Si tested under vacuum conditions for use in nanosatellites

**Bryanda G. Reyes-Tesillo**[1], **Genoveva Hernández-Padrón**[2]*, **Jorge A. Ferrer-Pérez**[3], **Alfredo Maciel-Cerda**[4]

1 Posgrado en Ciencia e Ingeniería de Materiales, Centro de Física Aplicada y Tecnología Avanzada, Universidad Nacional Autónoma de México Campus Juriquilla Querétaro, Querétaro, México,
2 Departamento de Nanomateriales, Centro de Física Aplicada y Tecnología Avanzada, Universidad Nacional Autónoma de México Campus Juriquilla Querétaro, Querétaro, México, 3 Departamento de Ingeniería Aeroespacial, Unidad de Alta Tecnología-Facultad de Ingeniería, Universidad Nacional Autónoma de México Campus Juriquilla Querétaro, Querétaro, México, 4 Departamento de Reología y Mecánica de Materiales, Instituto de Investigaciones en Materiales, Universidad Nacional Autónoma de México Ciudad Universitaria, Ciudad de México, México

* genoveva@fata.unam.mx

**Data Availability Statement:** All relevant data are within the manuscript and its Supporting information files.

## Abstract

A hybrid coating made of poly (methyl methacrylate) with $SiO_2$-$TiO_2$ particles (PMMA/$SiO_2$-$TiO_2$) has been developed for use as a coating on nanosatellites, evaluating its resistance to high vacuum by quantifying its weight loss. The coating was applied on an Al 7075 aluminum substrate used for the aerospace sector. PMMA/$SiO_2$-$TiO_2$ hybrid coatings were prepared using sol-gel reaction in situ assisted with sonochemistry. The $SiO_2$ particles and $TiO_2$ (50:50% wt. of rutile/anatase) particles by tetraethyl orthosilicate (TEOS), and Titanium (IV) Isopropoxide (TIPO). Radical polymerization of methyl methacrylate (MMA) monomer was conducted with (3-mercaptopropyl) trimethoxy silane (3-MPTS) used as a coupling molecular agent, and benzoyl peroxide as a catalyst. The coatings obtained have a thickness of 20 μm which were deposited by blade coating technique on the substrate, obtaining homogeneous and defect-free coatings. Adhesion and hardness were measured using ASTM standards required for this sector. To evaluate its resistance to ultra-high vacuum, it was done as close as possible to the ASTM E-595 norm [1], where it indicates that the coatings must be evaluated at vacuum conditions of $10^{-5}$ Torr and 125°C for a period of 24 hours. The coatings were evaluated before and after the test by spectroscopy analysis to determine a possible degradation in the chemical structure. The resulting weight loss not exceeding 0.02%, and the addition of Ti-O-Si particles led to an increase in chemical stability under vacuum conditions without affecting the chemical structure of the highly cross-linked PMMA/Ti-O-Si matrix, which was monitored by FTIR and Raman spectroscopy.

**Funding:** The author(s) received no specific funding for this work.

**Competing interests:** The authors declare no competing interests.

## Introduction

In the context of the aerospace sector the use of hybrid materials organics-inorganics derived from sol-gel techniques has emerged as promising avenue. These materials, characterized by combining two different phases with a homogeneous dispersion, offer a practical solution for enhancing the properties of aerospace materials. The organic phase, typically a polymer, can modify the mechanical properties, porosity, hydrophobicity, and flexibility, while the inorganic phase, often a metal oxide, reinforces the organic phase by increasing its hardness, mechanical stress and thermal properties. This unique combination of properties holds significant potential for the development of advanced aerospace materials, with direct implications for the industry [2].

The development of hybrid materials has allowed their application as coatings in the form of homogeneous protective films on already manufactured parts, providing anticorrosive, anti-icing, and hydrophobic properties [3–8], radiation resistance, erosion resistance, thermal-mechanical protection and prolonging the life of the material [2–4]. Due to their lighter weight and ability to adhere to various substrates, these materials have attracted great interest in space applications in the form of coatings to protect structural components, which has led to their use in nanosatellites.

Testing hybrid materials in ultra-high vacuum is essential to evaluate their performance emulating space-like conditions, as the lack of atmospheric pressure can affect the material, causing outgassing of the materials, altering their mechanical, electrical, and optical properties, to name a few, which places demands on the materials used in their manufacture. In the case of polymeric materials, high vacuum can cause surface contamination and loss of dimensional stability [5, 6], resulting in mass loss and cracking.

For the formation of hybrid coatings resistant to extreme environmental conditions, different materials for each of its phases have been used, in particular, a combination of polymers for the organic phase [9, 10], such as poly methyl methacrylate (PMMA) [11, 12], which thanks to its distinctive properties such as biocompatibility, low thermal conductivity, optical properties, etc., has been used. For the inorganic phase, titanium oxide ($TiO_2$) and silicon oxide ($SiO_2$) have been used for their properties such as hardness, low thermal conductivity, optical properties, vacuum resistance, low friction coefficient, etc. These materials have been synthesized in different [13], ways showing their efficient in different applications and different terrestrial environmental conditions, due to the final properties obtained [14–16].

This study focuses on the preparation and evaluation of PMMA/Ti-O-Si hybrid coatings using the in situ sol-gel technique assisted with sonochemistry to evaluate the improvement in the distribution of ceramic particles with ultrasound and to reduce the synthesis time, emulating ultra-high vacuum conditions and analyzing their adhesion and hardness by ASTM standards used for coatings used in this sector, in addition to monitoring their chemical structure by FTIR and Raman spectroscopy, which will allow us to determine the viability of these hybrid coatings and their resistance to this condition.

## Methodology

### Coating preparation PMMA/Ti-O-Si

For the formation of the hybrid materials, the coatings were synthesized by varying the molar ratio, the silicon oxide is formed using a precursor tetraethyl orthosilicate (TEOS) (Si $(OC_2H_5)_4$), titanium(IV) isopropoxide 97% (TYPE) ($C_{12}H_{28}O_4Ti$), 2-propanol (IPOH), ethyl alcohol (EtOH), nitric acid ($HNO_3$), and (3-mercaptopropyl)trimethoxy silane (3-MPTS) ($(3-SHC_3H_6)-SiOC_3H_9$), the monomer methyl methacrylate (MMA) ($CH_2 = C(CH_3)$

**Table 1. Molar ratio of hybrid coatings.**

| Sample | $SiO_2$ [mol] | $TiO_2$ [mol] | PMMA [mol] |
|:---:|:---:|:---:|:---:|
| **M1** | 0.02 | 0.02 | 1.17 |
| **M2** | 0.04 | 0.04 | 4.68 |

$COOCH_3$), benzoyl peroxide (PB) (($C_6H_5CO)_2O_2$). The concentrations were varied as can be seen in Table 1.

The hybrid coatings (M1 and M2) were synthesized *in situ* by the sol-gel technique supported by sonochemistry using the QSONICA 250 equipment at a frequency of 20 kHz with an amplitude at 80% and power of 50%.

It starts with a pre-hydrolysis to form $SiO_2$ using 0.04 mol TEOS, and 1 mol of EtOH is added in a constant dropwise manner in the sonotrode for 15 minutes, see Fig 1. Subsequently, 0.03 mol of TIPO is slowly added dropwise and left in the sonotrode for 5 min, and then add the $IPOH:H_2O:HNO_3$ solution with a molar ratio of 0.19:0.22:0.02 in the same manner and left for another 15 min, see Fig 2, thus continuing the hydrolysis and initiating condensation.

As there are two different phases (organic/inorganic), 0.04 mol of 3-MPTS is used as a coupling agent (CA), which allows us to create a bond between both phases. For the polymer matrix, the polymerization is performed using 0.18 mol of MMA and as precursors 0.0012 mol PB and finally 0.34 mol EtOH solvent is added, see Fig 3. The reaction is maintained for 15 minutes in the sonotrodo. With this procedure, it is proposed to form Si-OH, Si-OR, Ti-OR, and Ti-OH oligomers are produced, which condense to form $Ti-O_2$, $SiO_2$, and Ti-O-Si structures bonded through the coupling agent to PMMA.

The coatings have been applied to Al 7075 aerospace substrates using the blade coating technique, the drying of the coating does not use any type of curing, it is left at room temperature under normal conditions.

## Characterization techniques

Once the coatings were synthesized, they were characterized to evaluate that the *in-situ* sol-gel process was satisfactory using Raman spectroscopy, FTIR spectroscopy, scanning electron microscopy SEM and thermogravimetric analysis TGA. To evaluate the behavior of the chemical structure of the coating under vacuum conditions, monitoring is done using Raman and FTIR spectroscopy.

**Fig 1. Pre-hydrolysis for the formation of $SiO_2$.**

**Fig 2. Hydrolysis for the formation of TiO₂.**

For Raman spectroscopy, a Bruker model Senterra instrument was used with a 20x objective was used with a resolution of 9–15 cm$^{-1}$, an integration time of 8 s, 6 scans and a power source of 100 mW, the spectra were recorded in the range of 110–3400 cm$^{-1}$. The FTIR analysis was carried out on a Thermo Scientific model Nicolet 6700, using the ATR technique with crystal diamond, the spectra were recorded in the wave number range between 4000 and 400 cm$^{-1}$ with a resolution of 4 cm$^{-1}$. For SEM microscopy, the samples were observed in a HITACHI TM1000 scanning electron microscope at 15 KV with a backscatter detector, and for thermal degradation of hybrid was investigated by a thermogravimetric analyzer TA Instruments at room temperature to 800°C with a heating rate of 10°C under nitrogen atmosphere. The measurements were conducted using 10.5 mg samples.

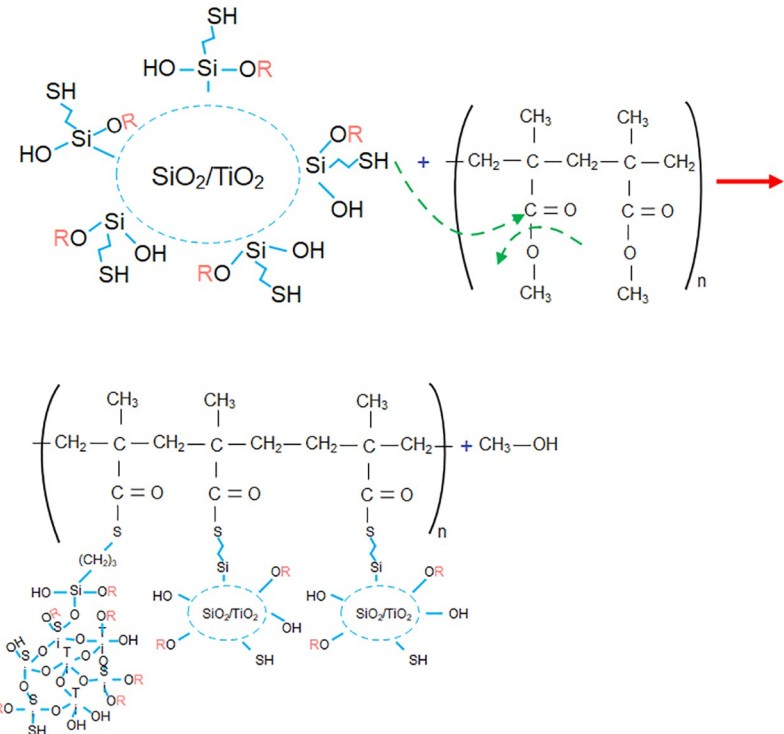

**Fig 3. Formation of PMMA/ Ti-O-Si hybrid.**

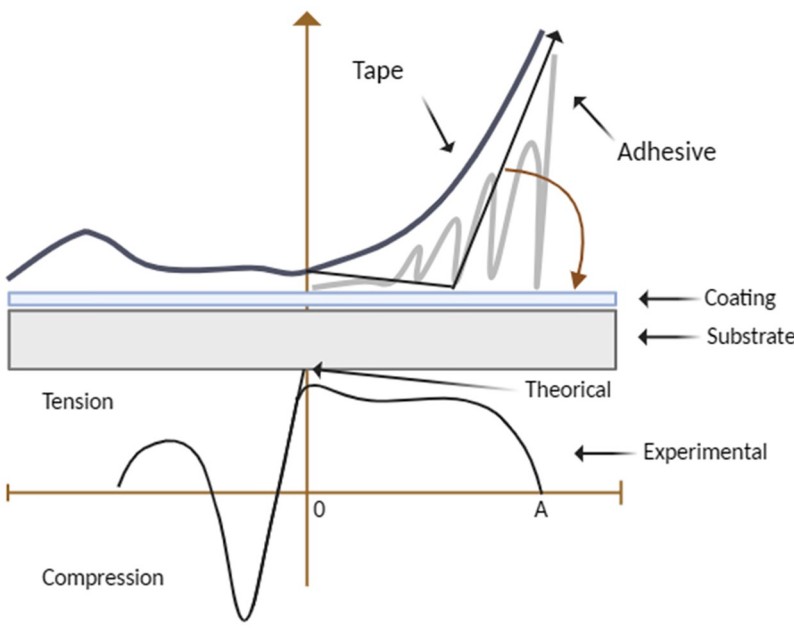

**Fig 4. Peeling profile.**

## Test of adherence and hardness

Coating adhesion is measured according to ASTM D 3359–17 [7], the standard used for organic/inorganic coatings in the automotive, aerospace, and other industries. Within the standard there are two methods: A and B. Method A is used for field tests and method B is used in the laboratory. In our case, we will use method B, for which a perpendicular grid pattern is made with six or eleven cuts in each direction and a sensitive tape is applied with pressure, and then removed by peeling as shown in Fig 4. This allows the adhesion of the coating to the substrate to be quantified, and it is classified according to the percentage of peel.

The hardness test was carried out according to ASTM D3363 norm [8], in an organic coating, transparent and pigmented. The test consists of evaluating the surface of the coating applying a constant force to try to generate a scratch at an angle of 45˚ with a pencil of known hardness based on the Wolff-Wilborn method [17]. The process should start with the pencil of lower hardness and go up until it leaves no mark, as shown in Fig 5.

## Ultra-high vacuum conditions test

The test was conducted by simulating conditions as close as possible to those specified in ASTM E-595 [1]. This gives us the humidity and temperature conditions of the room where

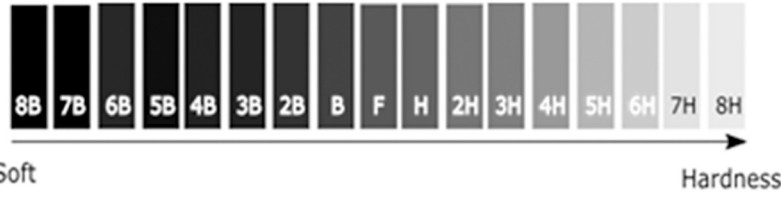

**Fig 5. Hardness scale.**

the coatings are weighed, the vacuum parameters in the chamber, and the calculation of the total mass loss (TML). The test was performed using an INTERCOVAMEX model 0000001, see Fig 6 thermal vacuum chamber to simulate these laboratory conditions.

The test lasts 24 hours under the mentioned conditions. At the end of this time, the chamber is cooled, and the vacuum is broken. The substrates are then reweighed in a semi-analytical balance. The weights were compared with those measured before the test to calculate the total mass loss (TML) using equation (Eq 1).

$$TML = \left( \frac{M_i - M_f}{M_i} \right) \cdot 100 \qquad (1)$$

*where*:

   *TML = Total mass loss* [%]
   $M_i$ = *Initial mass* [g]
   $M_f$ = *Final mass* [g]

## Results and discussion

### Ultra-high vacuum conditions test

The hybrid coatings obtained showed the following characteristics: uniformity, homogeneity, semi-transparency, and no cracking on the substrate, as shown in Fig 7.

The dispersive Raman spectrum in Fig 8 can be observed the presence of bands around 1700 cm$^{-1}$ belonging to the carboxylic groups (C = O), and a broad band from 2800–3000 cm$^{-1}$ corresponding to the -CH$_2$ and -CH$_3$ groups, characteristic bands of PMMA. In the range from 130 cm$^{-1}$ to 640 cm$^{-1}$, bands were detected indicating the presence of titanium oxide (Ti-O). In the case of sample M1, the absorption band at 993 cm$^{-1}$ and in sample M2 at 1006 cm$^{-1}$, indicating the presence of Si-OH. The interaction between titanium oxide and silicon oxide is identified at 1104 cm$^{-1}$ in sample M1 and at 1110 cm$^{-1}$ in sample M2 associated with the Ti-O-Si bond. The band at 2557 cm$^{-1}$ indicates the presence of the thiol group (-SH) of the coupling agent.

We confirmed, by FTIR spectroscopy in Fig 9, in both synthesized materials the formation of PMMA identified by the band at 2940 cm$^{-1}$ corresponding to the hydroxyl group, while in the region 3100–2800 cm$^{-1}$ the methyl groups are observed. The carboxylic acid shows two strongly coupled bonds, C = O and C-C-O, at 1728 cm$^{-1}$ and 1242 cm$^{-1}$ [18–20]. Ti-O-Si was found at 921 cm$^{-1}$ in sample M1 and in M2 it is found at 900 cm$^{-1}$, as a result of the reaction is presented the characteristic band in stretching of the Si-O-Si network is obtained in the range of 1091 cm$^{-1}$ and 800 cm$^{-1}$, Si-OH groups are found around 960 cm$^{-1}$ and 910 cm$^{-1}$. On the titanium side, Ti-O-Ti groups were identified at 700 cm$^{-1}$ and 750 cm$^{-1}$, while Ti-OH groups were observed at 584 cm$^{-1}$ [3, 21].

In the thermograms of materials M1 and M2 shown in Fig 10, we can see the thermal degradation process of PMMA [22, 23] with the inorganic components. The degradation temperature of the hybrids is higher with respect to the polymeric matrix, due to the crosslinking of the inorganic phase of titanium and silicon, which provides greater thermal stability to the hybrid [15, 24]. Fig 11 shows a temperature at 378˚C and a drop at 428˚C which can be assumed to be due to the fact that sample has a 1:4 PMMA ratio with respect to the M1. This is also observed by Raman spectroscopy, see, since the intensities of the main groups present higher intensity of the hybrid, PMMA/Ti-O-Si. However, by SEM microscopy there is no phase separation in the hybrid, and it is not reflected in the other properties evaluated, so it does not affect the performance of the material.

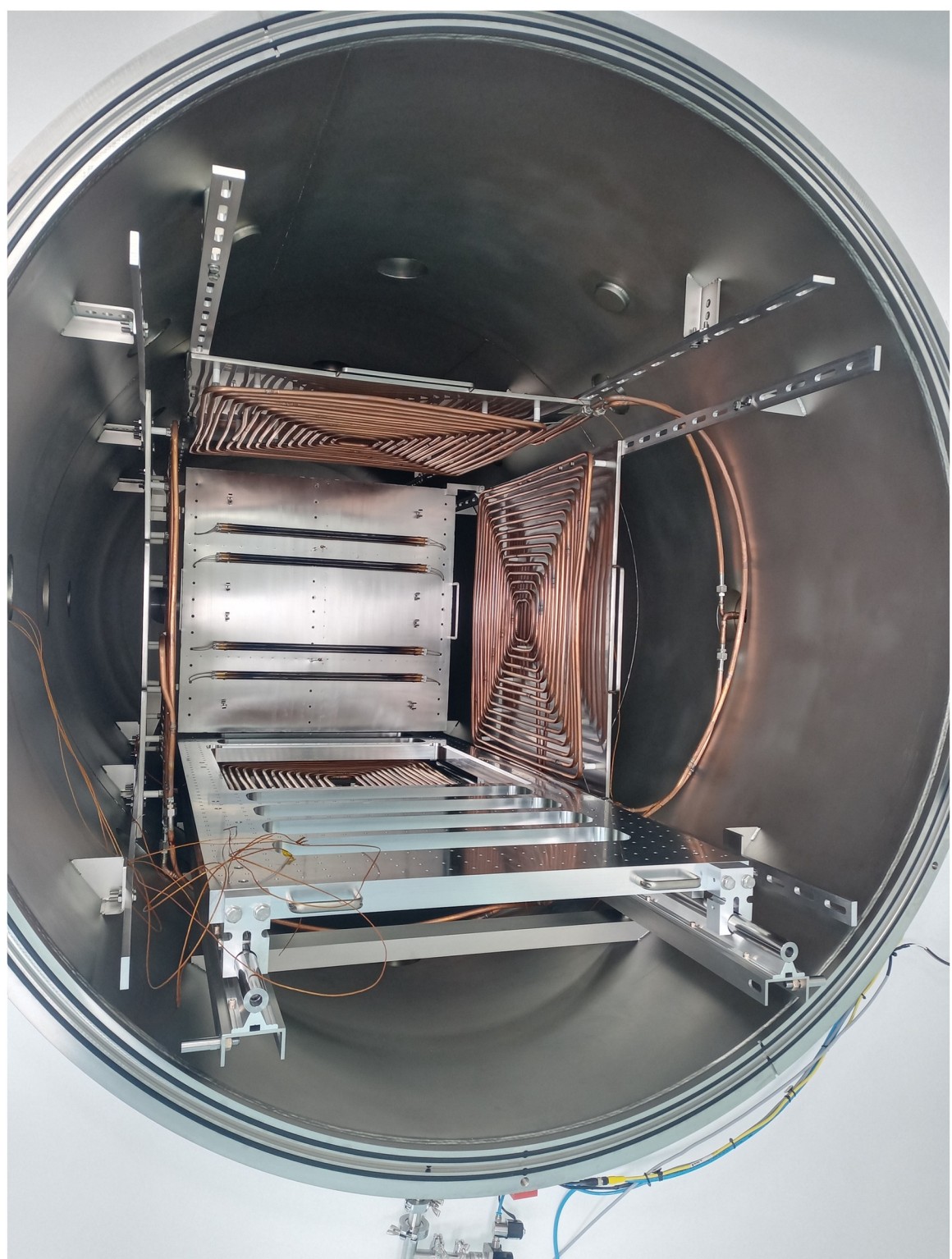

**Fig 6. Vacuum chamber.**

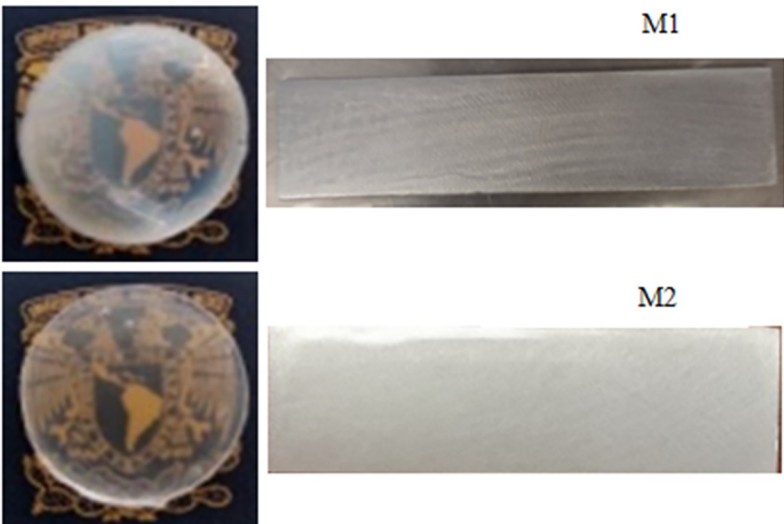

**Fig 7. Characteristics hybrid materials.**

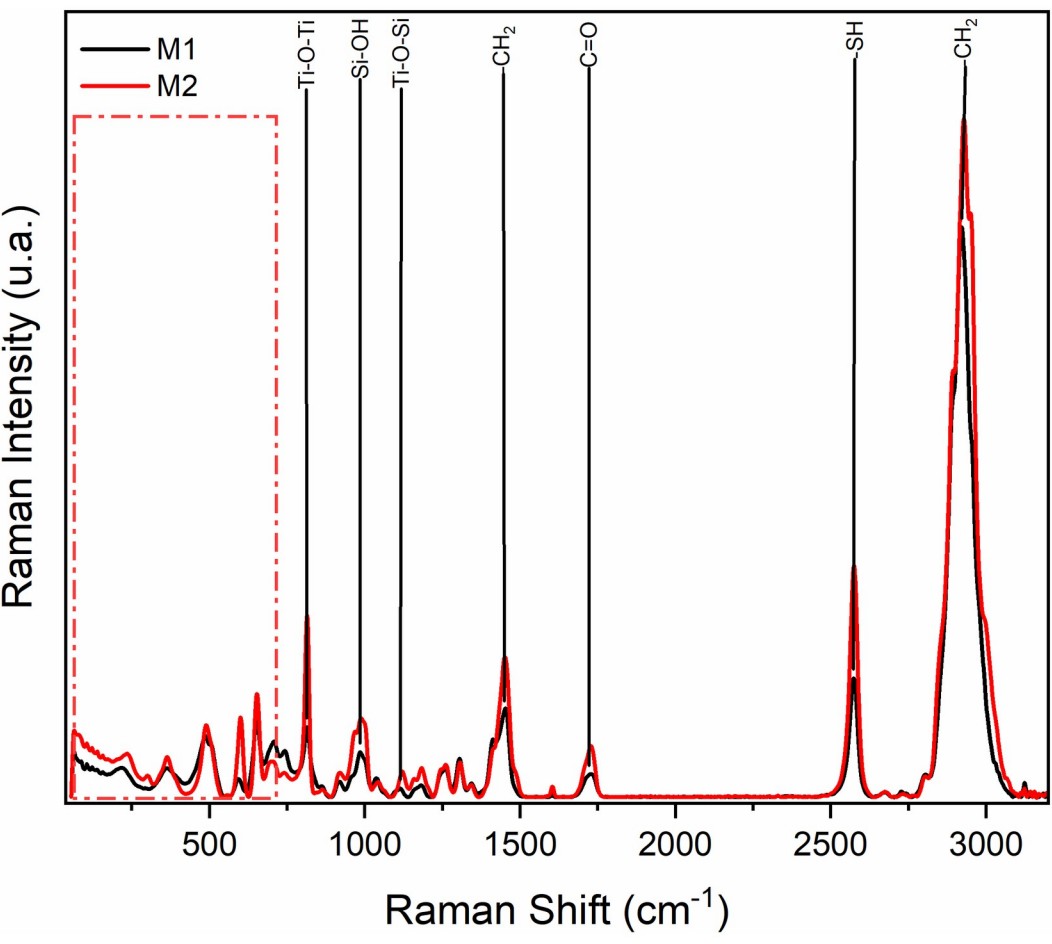

**Fig 8. Dispersive Raman spectra of PMMA/Ti-O-Si hybrid coating.**

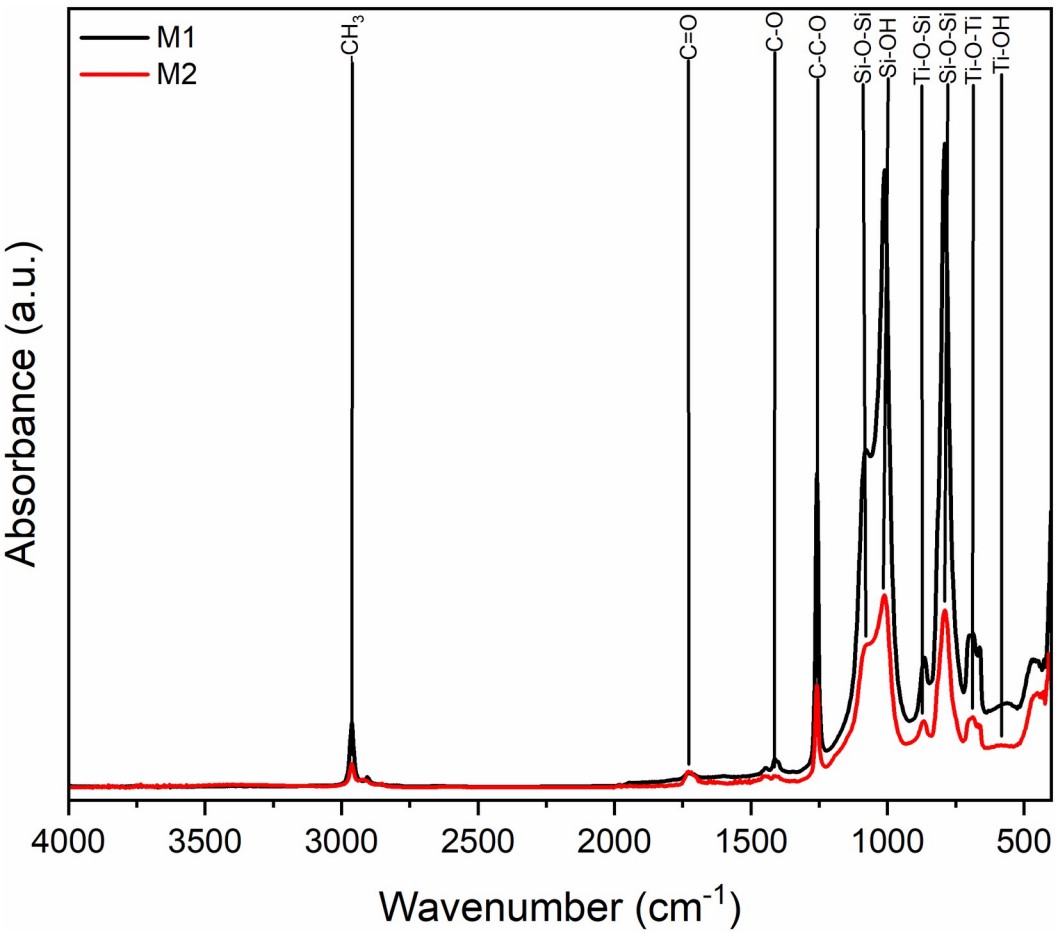

**Fig 9. Dispersive FTIR spectra of PMMA/Ti-O-Si hybrid coating.**

Figs 12 and 13 shows the scanning electron microscope (SEM) morphology of materials M1 and M2 respectively, where we can see that there is no phase separation. We can also see that the Ti-O-Si is homogeneously embedded in the PMMA matrix. Fig 14, corresponding to the Ti-O-Si composite, shows the morphology of the crystalline powder of the rutile-anatase phases (50:50%), previously characterized by RX and Raman.

After the test their adhesion and hardness properties were analyzed looking for any deterioration, in the M1 coating the adhesion had a percentage of detachment < 5% and in the scale of hardness it has a value of 4H, the M2 coating had a percentage of detachment of 0% and a hardness of 4H.

In both cases no affectation or any modification in the initial properties is observed, which could indicate some modification in the siloxane group (Si-OH) which is the one that provides the adhesion to the substrate.

## Weight loss evaluation

The samples were first dried for 24 hours at 23˚C and weighed at 20˚C and 40% RH. Once weighed, they are placed in the chamber on a rack, separated from each other to avoid cross-contamination, as shown in Fig 15; a sensor is placed near each plate to monitor the temperature in real-time.

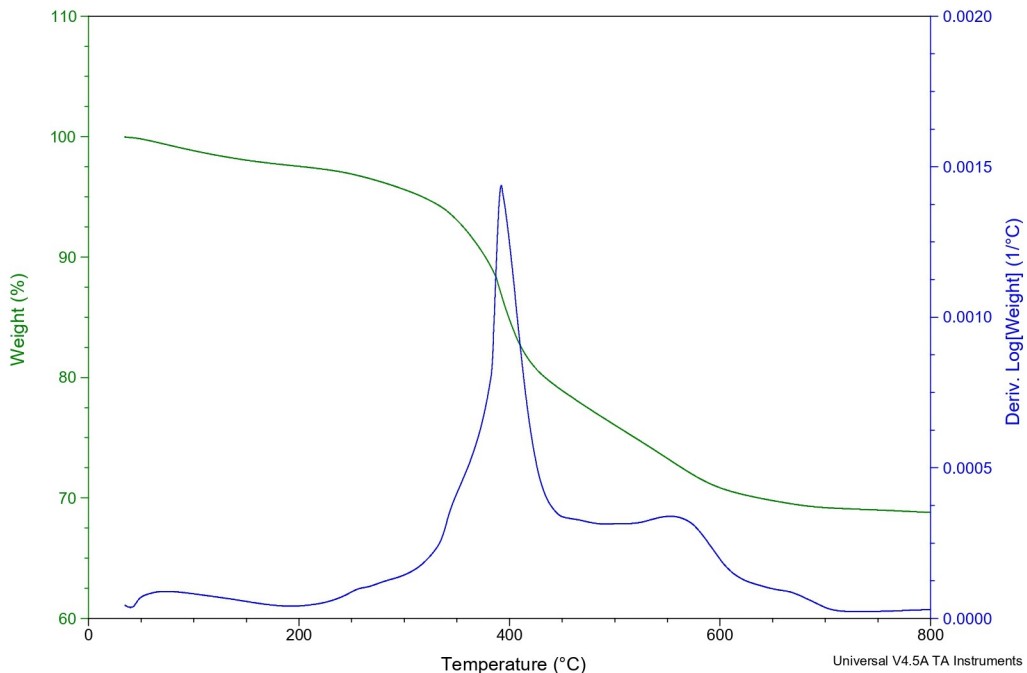

**Fig 10. Thermogravimetry of hybrid M1.**

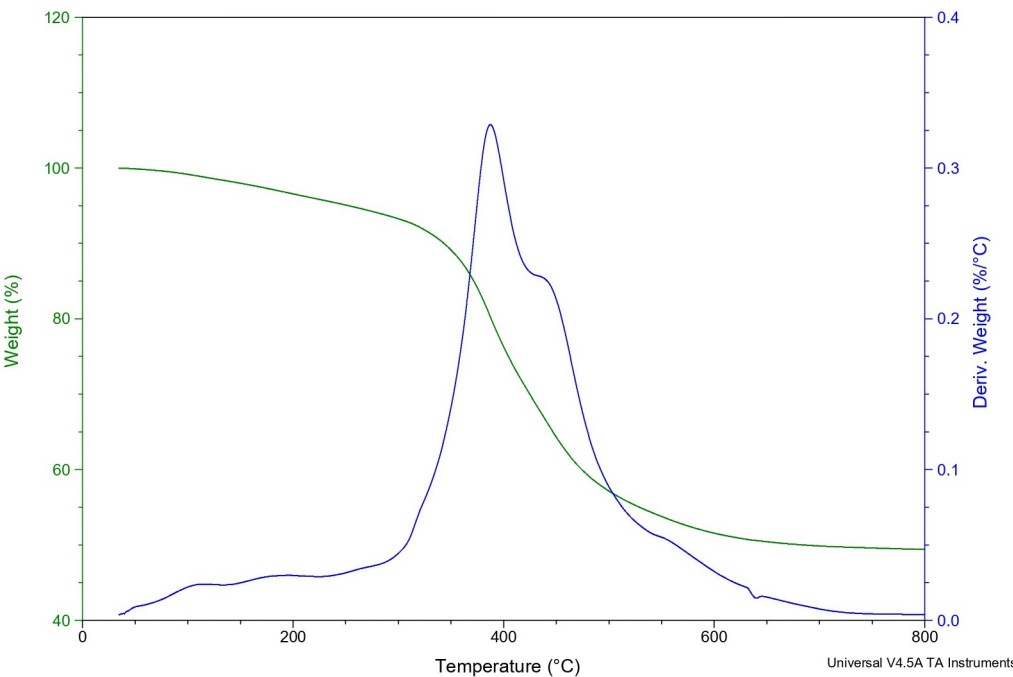

**Fig 11. Thermogravimetry of hybrid M2.**

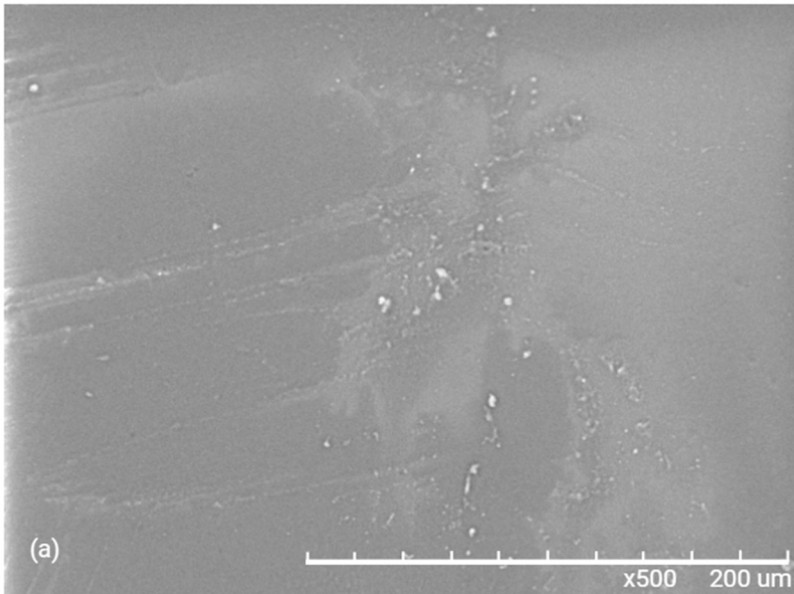

**Fig 12.  SEM image of M1 spectrum of elemental analysis.**

The vacuum chamber was brought to a vacuum of $10^{-5}$ Torr vacuum using a turbomolecular pumping system. The temperature inside the chamber was 125°C, and it was left for 24 h. Once the test was finished, the samples were weighed under conditions of 20°C and 36% RH.

Table 2 shows the average of each sample's initial and final weight and calculates the lost weight. The coating with the most significant total mass loss is M2, with 0.0336 g, while only M1 has a loss of 0.0192 g.

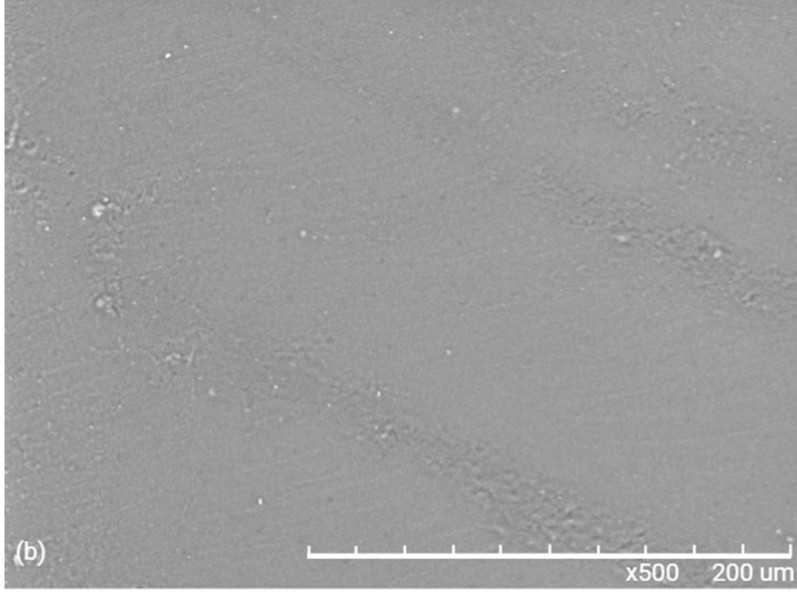

**Fig 13.  SEM image of M2 spectrum of elemental analysis.**

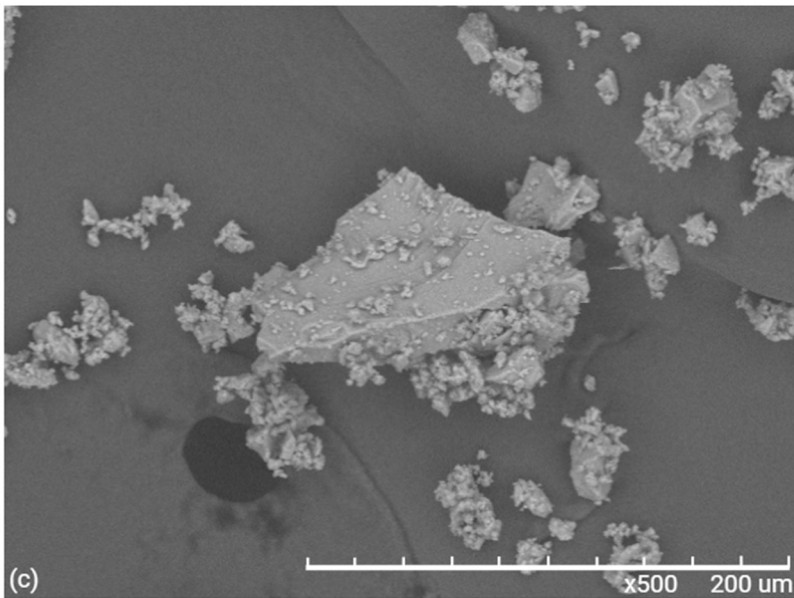

**Fig 14. SEM image of Ti-O-Si spectrum of elemental analysis.**

Applying Eq (1) and calculating the percentage of total mass loss was calculated. Both coatings do not have a TML value greater than 1%, which, according to the standard, means these coatings could still be tested in the different atmospheric conditions.

## Raman and FT IR analysis

The Raman spectra of M1 and M2 are displayed in Fig 16. The absorption bands around 1700 cm$^{-1}$ are due to the presence of carboxyl groups (C = O) and a broad band from 2800 to 3000 cm$^{-1}$ corresponding to the -CH$_2$ and -CH$_3$ groups, characteristic bands of the PMMA. In the

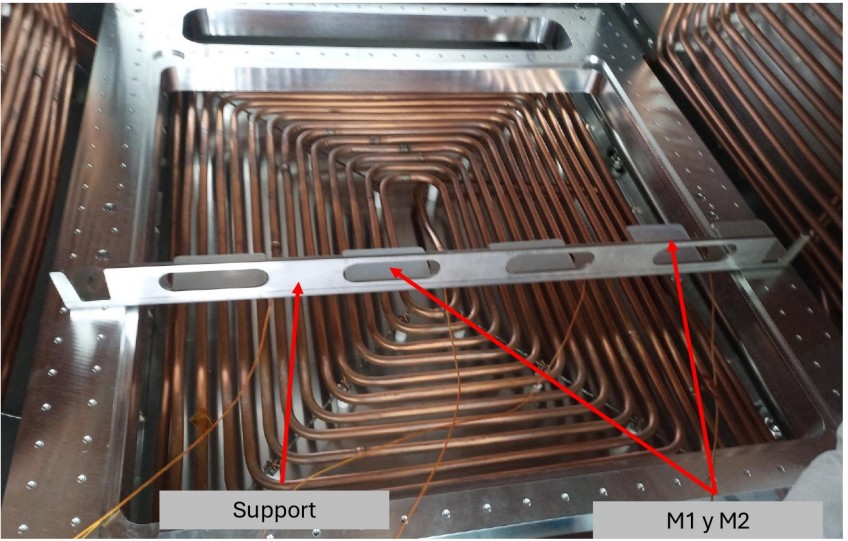

**Fig 15. Distribution of samples on the support.**

**Table 2. Weight of the coatings before and after the test showing the weight loss (TML).**

| Coatings | $\bar{M}_1$ [g] | $\bar{M}_2$ [g] | $\bar{M}_1 - \bar{M}_2$ [g] | TML [%] |
|---|---|---|---|---|
| M1 | 176.7694 | 176.7502 | 0.0192 | 0.011 |
| M2 | 177.7625 | 177.7289 | 0.0336 | 0.019 |

region from 130 to 640 cm$^{-1}$ the bands were detected indicating the presence of titanium oxide (Ti-O). In both spectra, the absorption band indicating the presence of Si-OH is observed at 980 cm$^{-1}$ in sample M1 and at 994 cm$^{-1}$ in sample M2. The interaction between titanium oxide and silicon oxide is identified at 1100 cm$^{-1}$ in M1 and at 1100 cm$^{-1}$ in M2, associated with the Ti-O-Si bond. The band at 2560 cm$^{-1}$ in M1 and at 2574 cm$^{-1}$ in M2 indicates the presence of the thiol group (-SH) of the coupling agent.

After the ultra-high vacuum test, a slight decrease in the intensity of the bands is observed in both coatings. It is possible to observe that in some samples there is a decrease in the methyl groups belonging to PMMA due to a possible degradation of the polymer matrix. On the other hand, the characteristic band of the Ti-O-Si bond does not decrease in intensity, which shows that the inorganic phase gives the chemical resistance to the hybrid material subjected to these ultra-high vacuum conditions.

These results were corroborated by FTIR spectroscopy show in Fig 17 spectra of samples M1 and M2 respectively. In both cases the characteristics absorption bands of PMMA were identified; around 3340 cm$^{-1}$ corresponding to the hydroxyl group. In the range 3100–2800 cm$^{-1}$ region the bands correspond to the methyl groups. The carboxylic acid presents two strongly coupled bonds, C = O and C-C-O, in the 1728 cm$^{-1}$ and 1242 cm$^{-1}$, characteristics bands of PMMA are located [25–27]. The sol-gel process was satisfactory, which is demonstrated by the presence of the absorption bands corresponding to the Ti-O-Si bonds between 990 cm$^{-1}$ and 910 cm$^{-1}$. As a result of the reaction, the characteristic band of the Si-O-Si network is obtained in the interval of 1091 cm$^{-1}$ and 800 cm$^{-1}$. The Si-OH groups are found around 960 and 910 cm$^{-1}$; on the titanium side, Ti-O-Ti groups have been identified at 700 cm$^{-1}$ and 750 cm$^{-1}$, while Ti-OH groups were observed at 584 cm$^{-1}$ [2, 3].

After exposure to ultra-high vacuum conditions, a significant change in the intensities of some of the previously mentioned absorption bands were observed, especially the hydroxyl (-OH) band, which is attributed to the degassing process that takes place under these conditions as a reaction of the material. The decrease in the intensities of the bands related to the Ti-O-Si and Si-O groups is most likely the result of a possible degradation of the material, which could be due to its degassing, but there is no greater in the hydroxyl group band. On the other hand, the PMMA absorption bands did not decrease in intensity.

## Conclusion

The hybrids were successfully synthesized by the sol-gel *in situ* processing of silica-titanium particles in the presence of the PMMA matrix assisted by sonochemistry. The use of sonochemistry allowed to accelerate the initial chemical reactions, reducing the synthesis time compared to the other non-ultrasonic methods, having also a dispersion of Ti-O-Si particles homogeneously distributed within the polymeric matrix.

Therefore, these coatings exhibit remarkable transparency due to the sonochemical synthesis, which allows homogeneity of the chemical interaction of TiO$_2$-SiO$_2$ (Ti-O-Si) particles and their appropriate molar composition, which is not easy to achieve with other conventional

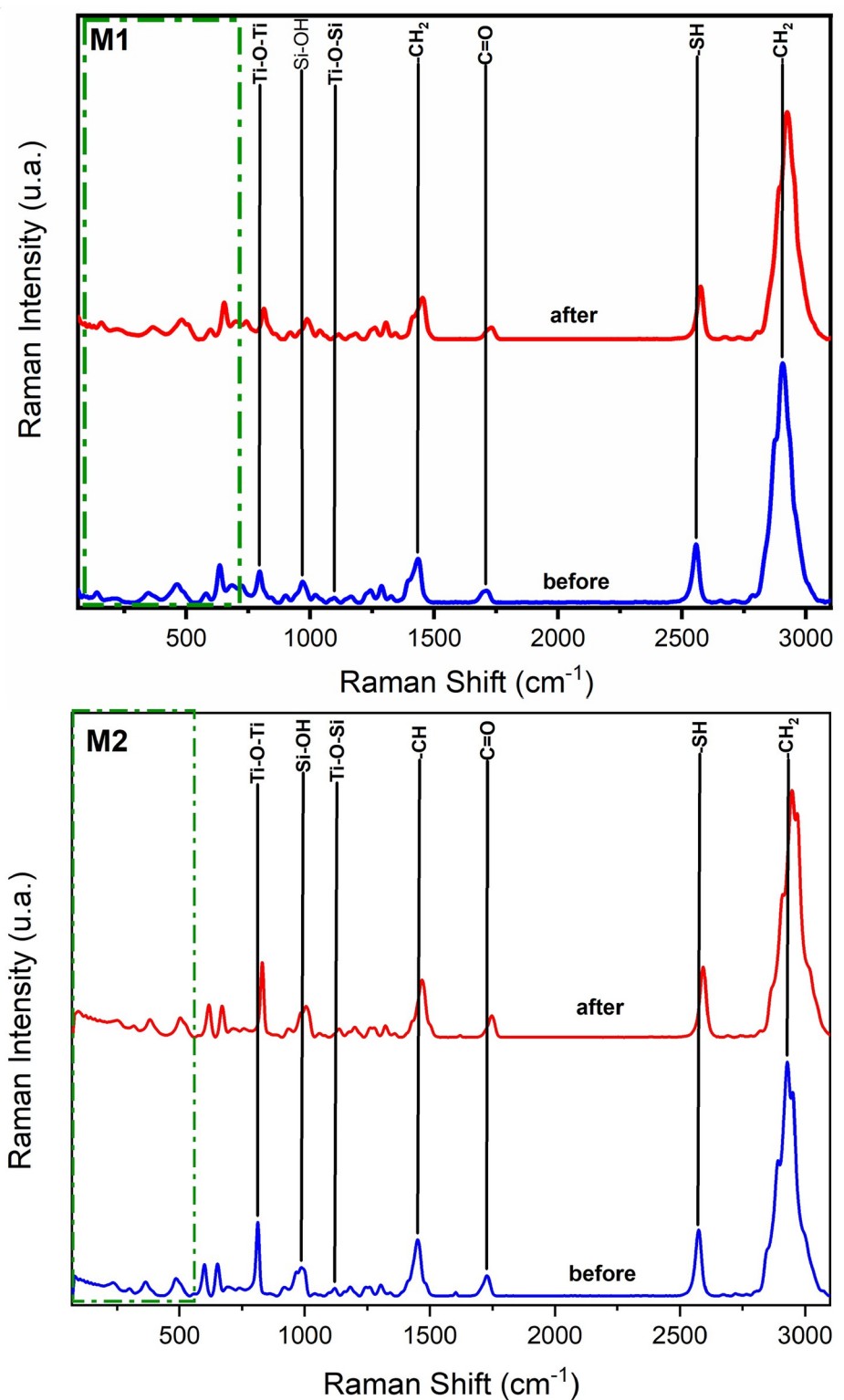

**Fig 16. Dispersive Raman spectra of PMMA/Ti-O-Si hybrid coating on the substrate.**

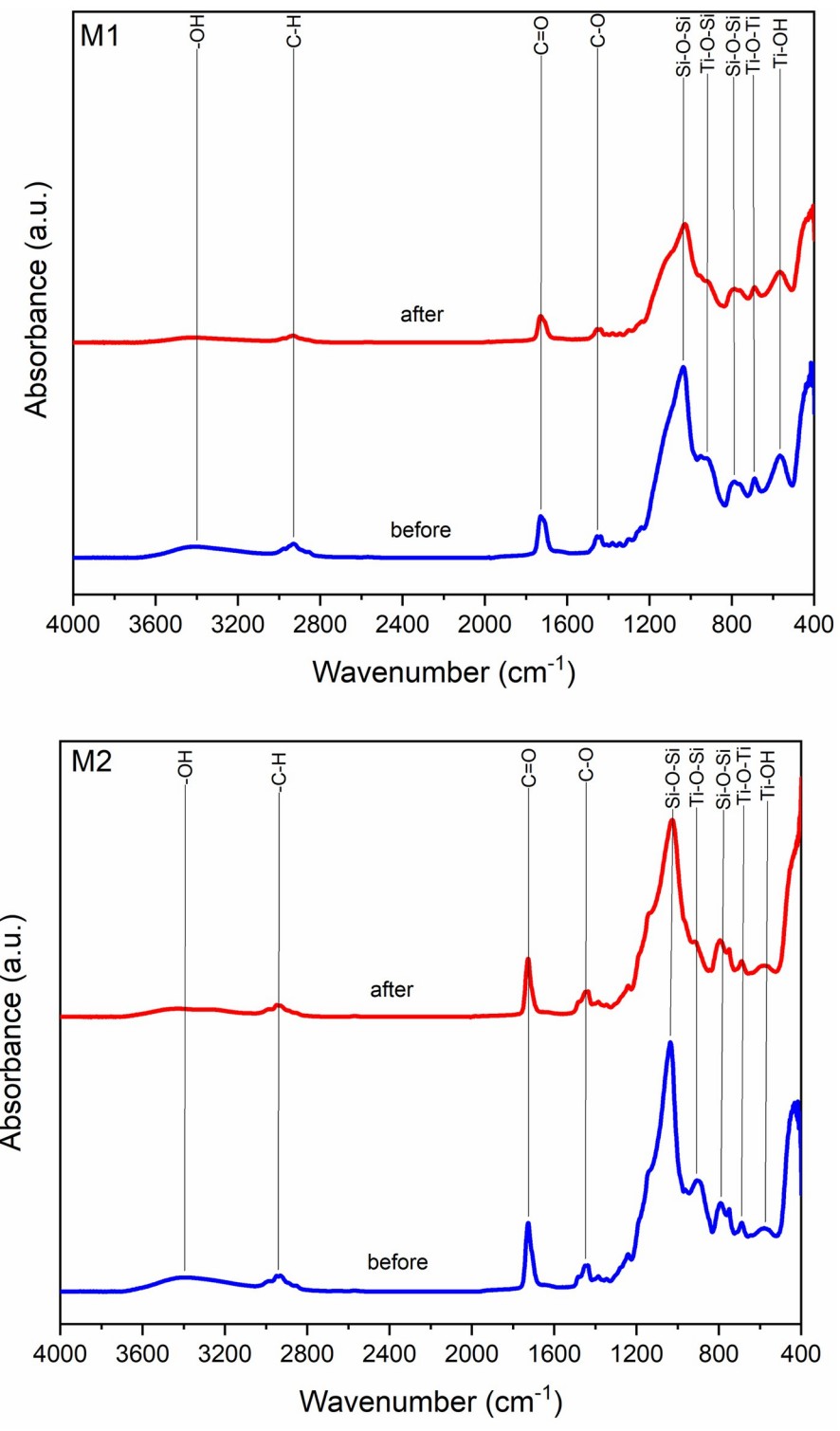

**Fig 17. FTIR spectra of PMMA/Ti-O-Si hybrid coating on the substrate.**

coatings. This has a significant influence on their morphological, thermal and mechanical properties.

The combination of vacuum endurance testing with Raman and FTIR spectroscopy techniques provides an in-depth and detailed evaluation of materials and components. It enables the detection of the chemical and structural changes that occur under vacuum conditions. This integrated approach is critical in the development of materials and products that must operate in low-pressure environments to ensure reliability and performance.

The prepared M1 and M2 coatings provide a reliable option for the aerospace industry because of the importance of ultra-high vacuum resistance of the coatings on Al 7075 substrates. After the high-vacuum test, their adhesion was significantly good, showing a peel percentage of no more than 5% and an average hardness of 4H, their weight loss was no more than 0.02%, which meets the established standard.

In general, the synthesized coatings did not show any significant chemical or mechanical degradation, which allows us to propose them as coatings in the aerospace sector.

Further studies of these hybrid materials we are related to other evaluations for a passive thermal control system and for analyzing their resistance to low temperatures. As well as their evaluation for other environmental conditions such as resistance to atomic oxygen attack, radiation resistance and to the effects of cosmic dust. The preparation these coatings would open a gap for their applications on other industrial sectors.

## Supporting information

**S1 File.**
(XLSX)

## Acknowledgments

The authors thank LaNCaM, LN-INGEA, and the Space Systems Integration Laboratory for the characterization infrastructure, Eriseth Reyes Morales of the Instituto de Investigaciones en Materiales, and Dra. Marina Vega González, technical support team SEM analysis Laboratorio de Geoquímica de Fluidos Corticales, Instituto de Geociencias, UNAM.

Bryanda Guadalupe Reyes Tesillo thanks to CONAHCYT for the scholarship for doctoral studies and the Programa Espacial Universitario (PEU) for facilitating the use of the specialized infrastructure.

## Author Contributions

**Formal analysis:** Bryanda G. Reyes-Tesillo, Genoveva Hernández-Padrón.

**Investigation:** Bryanda G. Reyes-Tesillo, Genoveva Hernández-Padrón, Jorge A. Ferrer-Pérez, Alfredo Maciel-Cerda.

**Methodology:** Bryanda G. Reyes-Tesillo, Genoveva Hernández-Padrón.

**Resources:** Genoveva Hernández-Padrón, Jorge A. Ferrer-Pérez.

**Supervision:** Genoveva Hernández-Padrón.

**Validation:** Genoveva Hernández-Padrón, Jorge A. Ferrer-Pérez, Alfredo Maciel-Cerda.

**Writing – original draft:** Bryanda G. Reyes-Tesillo, Genoveva Hernández-Padrón.

**Writing – review & editing:** Bryanda G. Reyes-Tesillo, Genoveva Hernández-Padrón, Jorge A. Ferrer-Pérez, Alfredo Maciel-Cerda.

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
