## [Decision Letter · Decision Letter 0]

6 Aug 2024

PONE-D-24-26920HYBRID COATING PREPARED WITH PMMA/Ti-O-Si TESTED UNDER VACUUM CONDITIONS FOR USE IN NANOSATELLITESPLOS ONE

Dear Dr. Hernández Padrón,

Thank you for submitting your manuscript to PLOS ONE. After careful consideration, we feel that it has merit but does not fully meet PLOS ONE’s publication criteria as it currently stands. Therefore, we invite you to submit a revised version of the manuscript that addresses the points raised during the review process.

We look forward to receiving your revised manuscript.

Kind regards,

Komal Rizwan, PhD

Academic Editor

PLOS ONE

Journal Requirements:

Reviewers' comments:

Reviewer's Responses to Questions

**Comments to the Author**

1. Is the manuscript technically sound, and do the data support the conclusions?

Reviewer #1: Yes

Reviewer #2: Yes

2. Has the statistical analysis been performed appropriately and rigorously? 

Reviewer #1: Yes

Reviewer #2: Yes

3. Have the authors made all data underlying the findings in their manuscript fully available?

Reviewer #1: Yes

Reviewer #2: Yes

4. Is the manuscript presented in an intelligible fashion and written in standard English?

Reviewer #1: Yes

Reviewer #2: Yes

5. Review Comments to the Author

**Reviewer #1:** This paper discusses a hybrid coating made of poly (methyl methacrylate) with SiO2-TiO2 particles (PMMA/SiO2-TiO2) developed for weight loss. I have some comments to modify the manuscript.

1. A paper has already been published in 2024 in which the researchers prepared the PMMA/SiO2-TiO2 composite. Is your synthetic protocol different from it? Please justify.

https://doi.org/10.1016/S0169-4332(03)00865-1

2. You can again check the reference style of the journal

3. How is the interaction between titanium oxide and silicon oxide identified in the Raman spectrum, and what is the wavenumber range for the Ti-O-Si bond?

4. What effects does the Raman spectrum's indication of the interaction between silicon oxide and titanium oxide have on the general characteristics and possible uses of the composite material?

5. The figures' quality is very poor. Kindly improve it. As you can see in the figure of SEM and figure 13, and 14, even the digits are not visible.

6. What is the novelty of your research paper? Kindly discuss it briefly in the conclusion section.

I suggest a major revision on the base of these comments.

**Reviewer #2:** The article "Hybrid Coating Prepared with PMMA/Ti-O-Si Tested Under Vacuum Conditions for Use in Nanosatellites" investigates the development and performance of a PMMA/SiO2-TiO2 hybrid coating applied to Al 7075 alloys, crucial in aerospace applications. The hybrid coatings, synthesized via sol-gel reaction assisted with sonochemistry, integrate SiO2 and TiO2 particles (rutile/anatase mix) via TEOS and TIPO precursors. Radical polymerization of MMA monomer with 3-MPTS coupling agent and benzoyl peroxide catalyst ensures a homogeneous, defect-free coating with strong adhesion and a hardness of 4 H, approximately 20 µm thick. Addressing the bellow questions would strengthen the article's impact and broaden its applicability in advanced material sciences.

1. The article claims minimal weight loss (0.02%) and enhanced chemical stability under vacuum (10^-5 Torr, 125 °C). How was this stability quantitatively measured, and were there comparative studies with other coatings or conditions?

2. Authors need to edit their research work so that it is clearer what the study's topic and methodology are. Authors may cite, for example, literature outlining the interest in recent study results released by researchers to bolster their introduction part.

https://doi.org/10.3390/pr11030959, https://doi.org/10.1016/j.surfcoat.2024.131128, https://doi.org/10.1016/j.scriptamat.2023.115763, https://doi.org/10.1016/j.jmst.2023.05.082.

3. FTIR and Raman spectroscopy were used to monitor the chemical structure before and after vacuum testing. Could the authors elaborate on specific changes observed in the spectra and correlate them with the coating's performance metrics?

4. Has the durability of the PMMA/SiO2-TiO2 coating been assessed over extended periods or under cyclic vacuum conditions that mimic operational scenarios for nanosatellites?

5. What are the challenges and considerations in scaling up the synthesis of these coatings for industrial applications, particularly in the context of large-scale production for aerospace industries?

6. Considering the promising results, what are the next steps in research or potential modifications to further enhance the performance or broaden the application scope of PMMA/SiO2-TiO2 coatings? .

7. The authors should compare their results with previous studies, which are included in a table of references below.

https://doi.org/10.1016/j.corsci.2023.111591, https://doi.org/10.1007/s40194-024-01704-w, doi: 10.1109/TNSE.2023.3342938, https://doi.org/10.1002/sus2.228

6. PLOS authors have the option to publish the peer review history of their article (what does this mean?). If published, this will include your full peer review and any attached files.

Reviewer #1: No

Reviewer #2: No

---

## [Author Response · Author response to Decision Letter 0]

25 Oct 2024

The authors would like to thank the reviewers and the editor for their time and contributions to improve this manuscript.

The file is attached: Response to Reviewers

---

## [Editor Report · Decision Letter 1]

31 Oct 2024

HYBRID COATING PREPARED WITH PMMA/Ti-O-Si TESTED UNDER VACUUM CONDITIONS FOR USE IN NANOSATELLITES

PONE-D-24-26920R1

Dear Dr.,

We’re pleased to inform you that your manuscript has been judged scientifically suitable for publication and will be formally accepted for publication once it meets all outstanding technical requirements.

Kind regards,

Komal Rizwan, PhD

Academic Editor

PLOS ONE
---

## [Editor Report · Acceptance letter]

5 Nov 2024

PONE-D-24-26920R1 

PLOS ONE

Dear Dr. Hernández-Padrón, 

I'm pleased to inform you that your manuscript has been deemed suitable for publication in PLOS ONE. Congratulations! Your manuscript is now being handed over to our production team.

Kind regards, 

on behalf of

Dr. Komal Rizwan 

Academic Editor

PLOS ONE